# Neural-Network-Based Ultrasonic Inspection of Offshore Coated Concrete Specimens

**Azamatjon Kakhramon ugli Malikov** [1], **Young H. Kim** [2], **Jin-Hak Yi** [3], **Jeongnam Kim** [1], **Jiannan Zhang** [1] **and Younho Cho** [4,*]

1 Graduate School of Mechanical Engineering, Pusan National University, Busan 46241, Korea; malikov@pusan.ac.kr (A.K.u.M.); kjnnoah@pusan.ac.kr (J.K.); zjn1997@pusan.ac.kr (J.Z.)
2 Institute of Nuclear Safety and Management, Pusan National University, Busan 46241, Korea; yhkim627@pusan.ac.kr
3 Coastal Development and Ocean Energy Research Center, Korean Institute of Ocean Science & Technology, Busan 49111, Korea; yijh@kiost.ac.kr
4 School of Mechanical Engineering, Pusan National University, Busan 46241, Korea
* Correspondence: mechcyh@pusan.ac.kr; Tel.: +82-51-510-2323

**Abstract:** A thin layer of protective coating material is applied on the surface of offshore concrete structures to prevent its degradation, thereby extending the useful life of the structures. The main reasons for the reduction in the protective capability of coating layers are loss of adhesion to concrete and flattening of the coating layer wall. Usually, the state of the coating layer is monitored in the setting of water immersion using ultrasonic inspection methods, and the method of inspection still needs improvement in terms of speed and accuracy. In this study, the ultrasonic pulse echo method was used in a water immersion test of the coated specimens, and continuous wavelet transform (CWT) with complex Morlet wavelets was implemented to define the received waveforms' time of flight and instantaneous center frequency. These allow one to evaluate the thickness of the coating layer during water immersion. Furthermore, phases of reflected echoes at CWT local peaks were computed using a combination of Hilbert transforms (HT) and wave parameters derived from CWT. In addition, three relative wave parameters of echoes were also used to train deep neural networks (DNN), including instantaneous center frequency ratio, CWT magnitude ratio, and phase difference. With the use of three relative waveform parameters of the DNN, the debonded layer detection accuracy of our method was 100%.

**Keywords:** coating material; pulse-echo; continuous wavelet transfer; Hilbert transform; deep neural networks





## 1. Introduction

Over the past few years, the demand for offshore concrete construction has increased significantly. It has been widely implemented in civil and military terminals, offshore airports, offshore wind power plants, sea lighthouses, offshore oil construction, etc. Concrete structures are exposed to both physical and chemical degradation offshore. The main chemical deterioration is caused by seawater constituents on cement hydration products and the crystallization of salts in the concrete [1,2]. Water waves, floating objects, freezing, and thawing cause physical erosion [3,4]. To ensure the durability of concrete structures, protective coating layers are applied to increase resistance to those negative factors.

A variety of internal and external stimuli can induce coating adhesion failure. Mechanical stress, environment-induced thermal stress, and corrosion are the most typical reasons for adhesion failure [5]. The delamination or blistering of the marine concrete structure is the main phenomenon leading to the deterioration of the coating layer. The blistering of the coating layer can occur during the application of the coating material due to the air trap. The coating layer can also deteriorate after application due to external factors [6,7].

The wall thinning of the coating layer and debonding of the coating layer from the concrete reduce protection capability, and in time, detection and repair procedures must be carried out [8,9]. Mainly ultrasonic methods are used for the monitoring of the structural health of coated concrete structures. Despite many achievements in ultrasonic nondestructive testing, the precise detection of defects relies on the experience and theoretical knowledge of the technicians.

The frequency of the ultrasonic beam plays an important role in the precise determination of the bonding delamination and the thickness of the coating layer [10,11]. The thin coating layer requires high frequencies; on the other hand, the ultrasonic losses increase with frequency [12,13]. Recently, a method for detecting metallic layer debonding using the resonant frequency of reflected ultrasonic echo waves has been implemented [14]. The limitation of the resonance-based bonding estimation is that the wavelength of the waveform is required to be an integer multiple of the thickness, which is almost impossible in the case of coating layers.

The acoustic mismatch between the coating material and concrete depends on the bonding strength between layers. The acoustic difference reaches its peak when the coating material is disbanded from the concrete specimen due to the presence of air between the two layers [15,16]. The phase of the reflected waveform can be used to estimate the acoustic properties of the material of the multilayered materials [17–19]. Additionally, Wang et al. [20] used several parameters of the chirp and experimentally found that the phase of the waveform was the most sensitive one among wave parameters. However, in the experiment, the air-coupled thermal wave radar imaging method was used to detect defects in the laminates. The wave phase can be estimated using the Hilbert transform, which is an efficient method with low computational requirements [21,22]. It allows the implementation of analytic waveforms to calculate the wave phase [23].

Recently, time-frequency analysis based on short-time Fourier transform of the waveform has been used to detect debonding of coating layers [24,25]. A major limitation of the short-time Fourier approach is its fixed-length time window, and accurate time and frequency resolution cannot be obtained simultaneously. On the other hand, continuous wavelet transform can provide a finer scale resolution compared to short-time Fourier spectral analysis. Furthermore, CWT (Continuous Wavelet Transform) is not limited to the fixed-length window function, and it yields a better resolution of the high-frequency components and also of the transient waveform [26–29]. Many other researchers have found that CWT has better resolution transient waveforms compared to STFT (Short-Time Fourier Transform) [30–32].

The ultrasonic detection of defects between two layers based on multiple parameters of the waveform and underwater inspection of coating layers increase the complexity of ultrasonic detection. In this study, a DNN (deep neural network) was implemented to ensure automatic detection of the debonded coating layer based on waveform parameters. Moreover, the DNN has been successfully implemented in many other fields and has shown good results in classification-related problems [33–35]. In the DNN's processes, hyperparameters and their values critically impact its performance and prediction accuracy. Because of a large number of hyperparameter values, manually adjusting hypermeters is a time-consuming process and requires a great deal of experience [36–38].

Until now, there have been few studies on the detection of the debonded section of the coating layer applied to the concrete structure in water immersion tests. Recent research has been based on the following methods: high-frequency wave echo decomposition [39], reflection coefficient theory [40,41], pulse-velocity-based methods [42,43], wave attenuation [44,45], the data fusion technique [46], the multifrequency method [47], and time-based parameters [48].

In this present research, the DNN-based automatic detection method for underwater coated concrete specimens is described. A novelty of the current research is the relative parameters of the waveform used to detect coated layer debonding, which eliminate the effect of variable parameters of reflected echoes on delamination detection. Both relative

and absolute parameters of the ultrasonic waveform, such as TOF, instantaneous center frequency, and the phase of the echoes were calculated using the CWT and HT of the waveform and used to train the DNN. The hyperparameters of the DNN were tuned by a Bayesian optimization method, and the results showed 100% accuracy in detection of debonded layers.

## 2. Theoretical Backgrounds

In the proposed debonding detection method, the measured waveform features were used to predict the bonding state of the coating layers. For this purpose, ultrasonic immersion tests were conducted with coated concrete specimens, and details will be described in this section. Experimentally measured waveforms were analyzed with CWT and HT, where the center of frequency, TOF, and phase of each echo were determined. In this section (Section 2), the theoretical details of the research are covered, and in the following section, the experimental setup is described.

### 2.1. Coated Concrete Material

The nondestructive method of coated concrete structural inspection is based on the estimation of variations in ultrasonic waveform parameters of the reflected echoes. When the layers detach, the waveform attenuation, signal to noise ratio, nonlinear parameter, phase of the echo, center frequency, time-of-flight, and other parameters of ultrasonic signals might change [49–52]. Usually, the ultrasonic inspection of the offshore coated structures is performed in underwater conditions. A schematic representation of the wave propagation path of the bonded coating layer is shown in Figure 1a, and the debonded condition is shown in Figure 1b. Due to the fast scattering of the wave in the concrete layer, the path of the ultrasonic wave is the same regardless of the bonding condition of the coating layer. The waveform reflection of the coated concrete structure is schematically shown in Figure 1, with A0 representing the reflected waveform from the coating material's surface, and A1 is the reflected waveform from the coating layer's back wall. The third echo in Figure 1 is denoted as A2, which corresponds to a twice reflected echo.

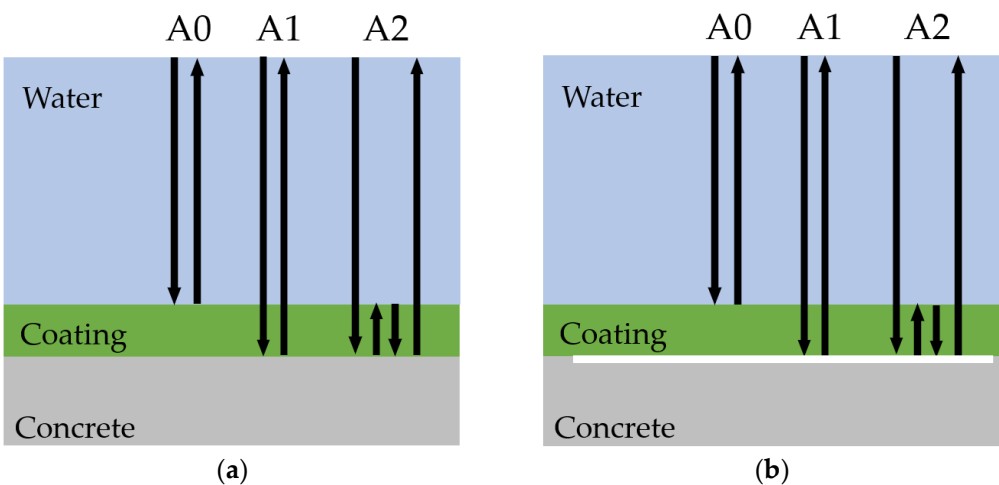

**Figure 1.** Schematic representation of the waveform propagation in a bonded coated layer (**a**) and a debonded coating layer (**b**).

In concrete coatings, the wave propagation is the same regardless of the bonding state of the layer (as shown in Figure 1), so it is very hard to determine the bonding state by analyzing the time-domain waveform. To identify bonded and debonded coating layers, waveform parameters are calculated from time domain waveforms. The CWT and HT of echo waves were used in this study to determine parameters such as attenuation rate, phase, and frequency. The following subsection describes the method.

### 2.2. Continuous Wavelet Transform

Two conventional algorithms were mainly used to compute the time-frequency spectrum of the waveform: STFT and CWT. The STFT requires the constant length of the window function, which will have a resolution limit of the high-frequency components of the waveform, and CWT allows this limitation to be overcome. In CWT, the waveform is multiplied by the mother wavelet and it can be written as [53]:

$$\mathrm{CWT}(a,b) = \frac{1}{\sqrt{a}} \int_{-\infty}^{\infty} y(t)\, \psi * \left(\frac{t-b}{a}\right) dt \tag{1}$$

where $y(t)$ is the examined waveform. The scale parameter $a\,(a > 0)$ defines time frequency resolution of the CWT [54,55] and the dilatation parameter $b \in \mathbb{R}$ of the CWT. In the CWT, the signal is divided by section and multiplied by the mother wavelet $\psi(t)$, where $*$ denotes the complex conjugate of the mother wavelet function. There are a significant number of wavelet types that can be selected as the mother wavelet function. There is not at present a universal method for the selection of the wavelets. We limit our research to using the complex Morlet wavelet [56,57], which can be written as:

$$\psi(t) = \frac{1}{\sqrt{\pi f_b}} e^{-(t^2/f_b)} e^{2i\pi f_c t} \tag{2}$$

where $f_c$ and $f_b$ are the central frequencies and they are bandwidth parameters of the Morlet wavelet. The central frequency parameter must be selected as $2\pi f_c > 5$, so it satisfies the admissibility condition. In this research, $2\pi f_c$ was set to 6 and $f_b = 0.5$ to ensure the admissibility condition for the function to have a zero mean and to be localized in both time and frequency space [58–60].

Based on the Heisenberg uncertainty principles, it is not possible to simultaneously obtain time and frequency resolution. The timescale resolution of the continuous wavelet transform is relevant. The scale parameter $a$ of the CWT is calculated by:

$$a = \frac{f_c f_s}{f_a} \tag{3}$$

where $f_s$ is the sampling frequency and $f_a$ is the central frequency of the wavelet corresponding to the scale $a$. Sampling frequency of the digital oscilloscope was set to 100 MHz. The finer scales produce maxima of magnitude, whereas coarse scales are not sensitive to noise. The advantages of the CWT allow the analysis of the signal time of arrival, bandwidth, and center frequency [61,62]. In the literature, the magnitude of the CWT is defined as [63]:

$$E(a,b) = |\mathrm{CWT}(a,b)|^2 \tag{4}$$

The local peaks of the CWT envelope magnitude correspond to the instantaneous center frequency and time of flight of the wave packets, correspondingly, in the frequency and time domains. The CWT magnitude reaches its maximum when $b = \tau_k$ and $f_a/a = f_k$ [63,64].

An experimentally acquired waveform is a real-valued time-varying signal. The analytic signal can be derived by applying HT to the experimentally measured waveform $y(t)$. The real part of the analytic complex signal $z(t)$ is equal to the original signal $y(t)$ [65,66]. The analytic complex signal can be expressed by:

$$z(t) = y(t) + iH(t) \tag{5}$$

where $H(t)$ is HT of the signal. The HT is mathematically expressed as the convolution of arbitrary signal defined with $1/t$, and it can emphasize the local properties of signal as follows [67]:

$$H(t) = \frac{1}{\pi} \int_{-\infty}^{+\infty} \frac{y(\tau)}{(t-\tau)} \tag{6}$$

The instantaneous phase $\phi_k$ of the reflected echoes can be estimated according to the following formula [68]:

$$\phi_k = \arctan\left(\frac{H(\tau_k)}{y(\tau_k)}\right) - 2\pi f_k(t - \tau_k) = \angle\theta_k - 2\pi f_k(t - \tau_k) \tag{7}$$

In this study, the phase of the reflected echoes $\phi_k$ was calculated from the instantaneous phase $\theta_k$ of the analytical signal. The instantaneous center frequency $f_k$ and time of flight ($\tau_k$) of the waveform were computed by CWT of the waveform and substituted in Equation (7). The center frequency and time of flight might also be found with HT; however, CWT with a complex wavelet allows one to precisely detect both two parameters even under a low signal-to-noise ratio.

*2.3. Waveform Parameters*

In the case of the coated concrete structures, due to the acoustic energy transfer from the coating layer of the concrete, the attenuation of the ultrasonic energy increases when the coating layer is bonded to concrete. The debonded layer possesses a relatively lower rate of attenuation because of the presence of air in the void between the coating material and concrete, since the ultrasonic energy from the back wall of the coating materials is reflected. Apart from this, the method of application of coating materials plays an important role; usually, the coating materials are applied by rollers, which cause the generation of the pores within the layer. The presence of the porosity in the coating layer causes variation in the attenuation rate. Moreover, the irregularities of the concrete surface lead to variations in coating layer thickness, which also cause variation in the waveform attenuation rate. As a consequence of the variations in the coating layer's properties and thickness, it is not possible to set a certain level of threshold of the attenuation rate for the bonding classification [69].

The reflection coefficient of the waveform also depends on the bonding state of coating materials. In order to calculate the reflection coefficient of the waveform reflected from two laminates with various bonding conditions, a massless spring model is employed because the stiffness of the spring model can be correlated with the bonding strength between layers. Table 1 below shows the expressions of reflection coefficient for three different bonding states between two layers, where $Z_1$ and $Z_2$ are acoustic impedances and $K$ corresponds to bonding stiffness between two layers [70,71]. Perfect bonding between the coating layer and concrete corresponds to ($K \to \infty$), with the reflection coefficient becoming only proportional to the acoustic impedances of adhesive and adherent interface, as shown in Table 1. On the other hand, when the coating layer is debonded from the concrete, the strength between coupled sections becomes equal to zero ($K = 0$), and the detached section is occupied by air and the reflection coefficient becomes equal to $-1$. This means that all waveform energy is reflected from the back wall of the coating layer, and the phase of the echo is reversed.

**Table 1.** Reflection coefficient of the waveform for bonded, debonded, and partial or imperfect bonding layer.

| Boding State | Formula |
|---|---|
| Perfect bonding | $R = \frac{Z_2 - Z_1}{Z_1 + Z_2}$ |
| Debonding | $R = -1$ |
| Imperfect Bonding | $R = \frac{Z_2 - Z_1 + i\omega\left(\frac{Z_1 Z_2}{K}\right)}{Z_1 + Z_2 + i\omega\left(\frac{Z_1 Z_2}{K}\right)}$ |

The aforementioned statements indicate that the first echo (assigned as A0 in Figure 1) from the surface of the coating layer should have the same reflection coefficient and phase regardless of the bonding condition of the coating with concrete. Furthermore, when the coating layer delaminates, the reflection coefficient of the second echo become negative since it reflected from the backwall and the phase of the echo will be changed by 180 degree. Additionally, the attenuation rate of the bonded coating layer will be faster than the debonded case. The next section describes how the theoretically derived outcomes were tested experimentally.

## 3. Experiments

### 3.1. Specimen Preparation

In order to verify the proposed theory, the coated concrete samples were fabricated. Initially, the concrete specimens with the dimension of 300 mm × 300 mm × 30 mm were cast in the laboratory condition. To avoid variation in the mechanical properties of the concrete specimen, the same batch of cement and aggregate was used for all concrete specimens. The cement content of the concrete mixture was 340 kg/m$^3$, whereas the water content was 83 kg/m$^3$ and the aggregate was a mix of fine and coarse pieces with a content 325 and 451 kg/m$^3$, respectively. The water reducer was added to the concrete mixture to increase the slump, and its content was 5 kg/m$^3$. The tensile strength of the fabricated concrete specimens was 5.1 MPa, which was measured 28 days after fabrication with three cylindrical samples of 100 mm diameter and 200 mm length [72].

The RS 500P coating material, which is produced by Chemco Company, was applied to concrete specimens. This type of coating material is mainly used for concrete, steel, and other materials. The applied coating material consists of two compounds consisting of epoxy resin and hardener. The weight ratio of resin to hardener was set at 5.1:1, and this ratio was suggested by the manufacturer. A properly weighted resin and hardener mixture was mechanically mixed for 5 min and then applied to the surface of the concrete specimen. Before the application of the coating material, the surface of the concrete specimen was blown by compressed air. This was done to remove dust and other material that might affect the bonding strength. Coating material density was measured as 1.67 g/cm$^3$ and wave propagation speed as 2.40 km per second.

To create the debonded part of the coating layer, a thin layer of polyethylene with a thickness of 0.05 mm and a size of 100 mm × 100 mm is placed in the center of the concrete specimen, and then a coating of material is applied to it, as shown schematically in Figure 2. In the rest of the concrete specimen, a coating material was directly applied to the surface of the concrete specimen, resulting in adhesion between the concrete and the coating. However, where a polyethylene layer is present, it avoids direct coupling between the coating material and the concrete. The acoustic impedance of the polyethylene layer is very close to that of the coating, so the thin film of the polyethylene layer will not affect the reflected echo signal.

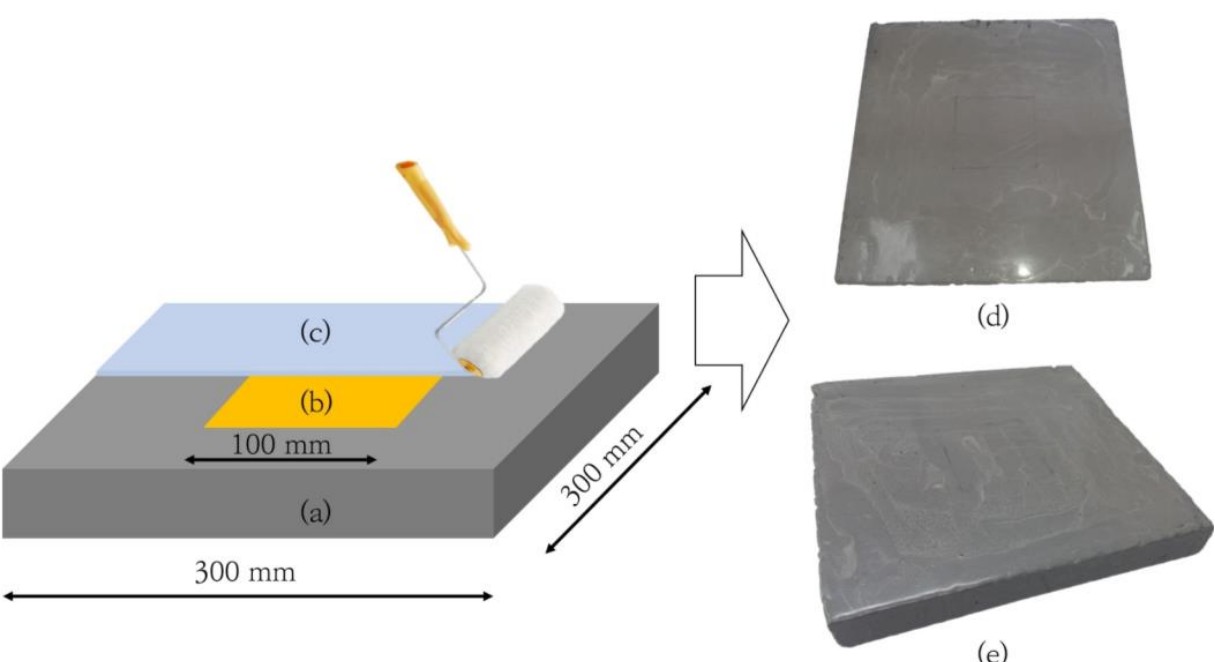

**Figure 2.** Specimen fabrication. (**a**) Concrete block, (**b**) thin polyethylene film, (**c**) coating layer, and (**d**,**e**) are the top- and side-view photos of the fabricated specimen.

Due to the non-adhesive property of polyethylene and the fact that it has compatible acoustic impedance with the coating layer, thin polyethylene layers can be used in ultrasonic nondestructive testing to simulate the delamination coating layer [73,74]. The coating layer thickness is determined by various properties and can range from very thin to 3 mm; however in this research, the optimal thickness design was not the main purpose. The thickness of coating layers in the fabricated specimens was less than 1 mm.

### 3.2. Immersion Scanning Device

For the ultrasonic immersion test, a spherically focused Krautkramer Alpha ultrasound probe with a center frequency of 3.5 MHz was implemented. The transducer was completely submerged in water at a focal length ($F_a$) from the specimen surface, as shown in Figure 3. The immersion setting was used for the nondestructive evaluation of the coated concrete plate, and the pulse-echo method was set as shown in the figure. The oscilloscope displays the waveform at a voltage scale and physically the voltage V measured by the transducer corresponds to the pressure P of the reflected wave. In the experiments, the digital oscilloscope (WaveRunner 604Zi, Teledyne LeCroy, Chestnut Ridge, NY, USA) with a sampling frequency of 40 GS/s was used to visualize and store signals from a pulser-receiver (5072 PR, Panametrics, Waltham, MA, USA).

In the experimental ultrasonic immersion test studies, tapered water was used, whereas in real-world applications, the offshore structures are located in seawater. Compared to tap water, seawater has a higher salinity and sediment concentration. This makes seawater denser, resulting in a slower propagation speed and faster attenuation rate of the ultrasonic energy [75,76]. The salinity and sediment content of the water do not affect the attenuation of ultrasonic energy in the coating material itself. Similarly, the difference in TOF of echoes from the surface and the back wall of the coating layer is not affected by the water's properties. Additionally, the phase change is independent of the salinity of the water. Wave propagation speed measured in tap water was 1.40 km per second, and the density was 1 g/cm$^3$.

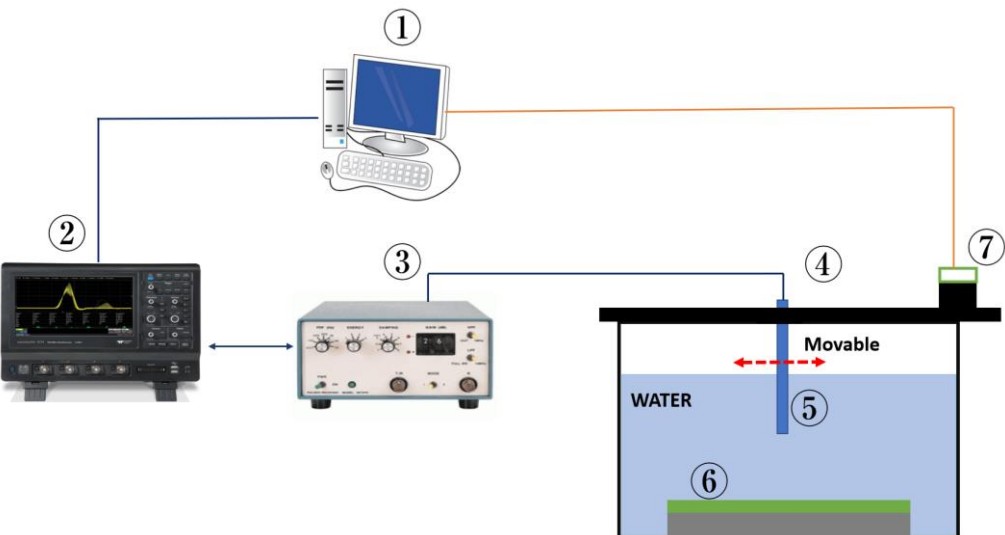

**Figure 3.** Instrumental setup: ① Computer, ② Lecroy Oscilloscope, ③ Pulser–receiver, ④ Ultrasonic probe, ⑤ Immersion scanning tank, ⑥ Coated concrete specimen, and ⑦ Probe positioning unit.

*3.3. Ultrasonic Probe*

The radiated acoustic sound can be concentrated by contact-focused transducers [77]. Immersion ultrasound testing uses a spherically focused transducer, in which an ultrasound beam passes through water coupled with solid interference. The focusing transducer has a focal length that varies with radius and curvature. In the interference between water and the coating layer, the ultrasonic beam is refracted, and the focal length shifts with the distance from the surface of the layer. Analogous to the near-field region of a plane-wave transducer, the focused transducer has an acoustic focal length $F_a$ that is quadratically proportional to the ratio of the radius of curvature $R_l$ to the transducer diameter. The acoustic focal length in water can be calculated by determining the geometry of the focused transducer from the formula below [78].

$$F_a = R_l \frac{1 - \frac{\left(\frac{\lambda}{2} - \frac{V_w}{V_l}\right)^2}{r^2 + h^2}}{\left(1 - \frac{V_w}{V_l}\right) + \frac{\lambda}{2h}} \tag{8}$$

$V_w$ and $V_l$ are the propagation speeds of waves in the water and in the ultrasonic probe lens, respectively. The geometrical parameters of an ultrasonic probe are $R_l$, $h$, and $r$, which are the curvature radius, depth, and radius of the lens, respectively. The wave speed in the water was 1.40 km per second, and the wavelength ("$\lambda$") of the wave in Equation (8) was 0.4 mm. At the focal length, the beam reaches its maximum intensity and minimum beam width [79]. By convention, the focal length is short compared to the near-field length of conventional planar transducers. The higher the frequency of the ultrasound beam, the subtler the defects that can be detected; however, the wave energy decays rapidly with increasing frequency. A 3.5 MHz focused transducer has been selected to prevent rapid attenuation of the ultrasonic energy and to provide adequate detection [80]. The beam profile affects the sensitivity of the transducer, and reducing the focal diameter reduces its penetration ability [81]. The ratio of the focal length of the focused transducer to the active (effective) diameter determines the degree (or quality) of the focusing property. The beam width $W$ at the focal point can be determined as follows [82]:

$$W = \frac{1.22\,\lambda}{2\,r} \tag{9}$$

As can be seen from Equation (9), the higher the center frequency of the probe, the narrower the beam width will be at a focal point. A large depth causes a decrease in signal-to-noise ratio (SNR) as the energy emitted from the transducer decreases beyond its focal point. Using focused probes, a maximum sensitivity can be achieved within a small, circular range within the bundle [83].

## 4. Experimental Results

The fabricated specimens were immersed in the water immersion test stand. An automated scanning system was used to obtain pulse-echo waveforms from the surface of the specimen. During ultrasound scanning of the specimen, the probe was shifted by 5 mm in each step, and corresponding coordinates of the probe were saved together with their accompanying reflected echo waveforms. Due to the manufacturing method of the concrete blocks, the surface roughness was not perfectly uniform. Therefore, the times of flight (TOFs) of the first echoes from the surface of specimen, A0, is not expected to be the same, due to the slight change in the distance between the transducer tip and the coated concrete surface. Additionally, the thickness of the applied coating layer is affected by the surface quality, and there may be slight variation in the coating layer thickness depending on the coating process. The ultrasonic scanning equipment allows quick scan of the surface of specimen and acquiring waveforms. The ultrasonic test results are shown in Figure 4a for the bonded coating and in Figure 4b for the debonded portion of the coating layer. From an initial glance at the obtained time domain waveforms, it is very difficult to find the distinguishable parameter.

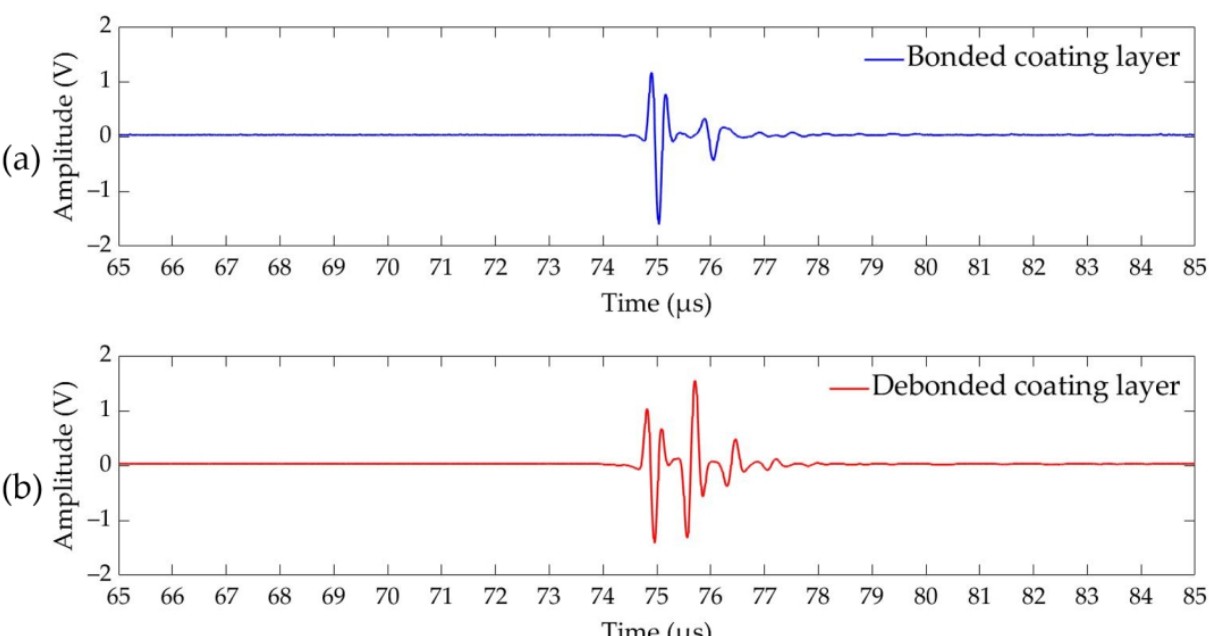

**Figure 4.** Received waveform from (**a**) bonded and (**b**) debonded coating layers.

As shown in Figure 4, the TOFs of the reflected echo from the surface of the coating layer (A0) and the wall of the coating layer (A1) were not identical. These differences in the TOF are due to the surface finishing of the concrete and the application of the coating, which does not produce a uniform layer thickness.

The CWT was applied to all waveforms to evaluate the ultrasound's characteristics. Figure 5a shows the waveform from the debonded section of the specimen, and its time–frequency representation is shown in Figure 5b. The local concentration in the contour plot corresponds to the maximum energy, and it defines the instantaneous center frequency of the echo. The time–frequency plot allows the instantaneous center frequency of each echo to be calculated separately. The corresponding center frequencies of the first and the

second reflection echoes were calculated, and the values are assigned, correspondingly, as "CF of A0" and "CF of A1" in Table 2. As we can see from the table, the instantaneous center frequencies of the A0 echoes, which were reflected from the surface of the coating layer, were close to the center frequency of the probe regardless of the bonding state of the coating layer. In fact, the values of the instantaneous center frequencies of the A0 echoes are related to the surface quality of the coating material. Whereas the second reflected echo (A1) arrives from the back wall of the debonding part, the waveform is scattered due to the interference between the coating layer and the concrete block. The center frequency of the A0 echo from the debonded layer was equal to 3.58 MHz, whereas the CF of A1 was 3.12 MHz.

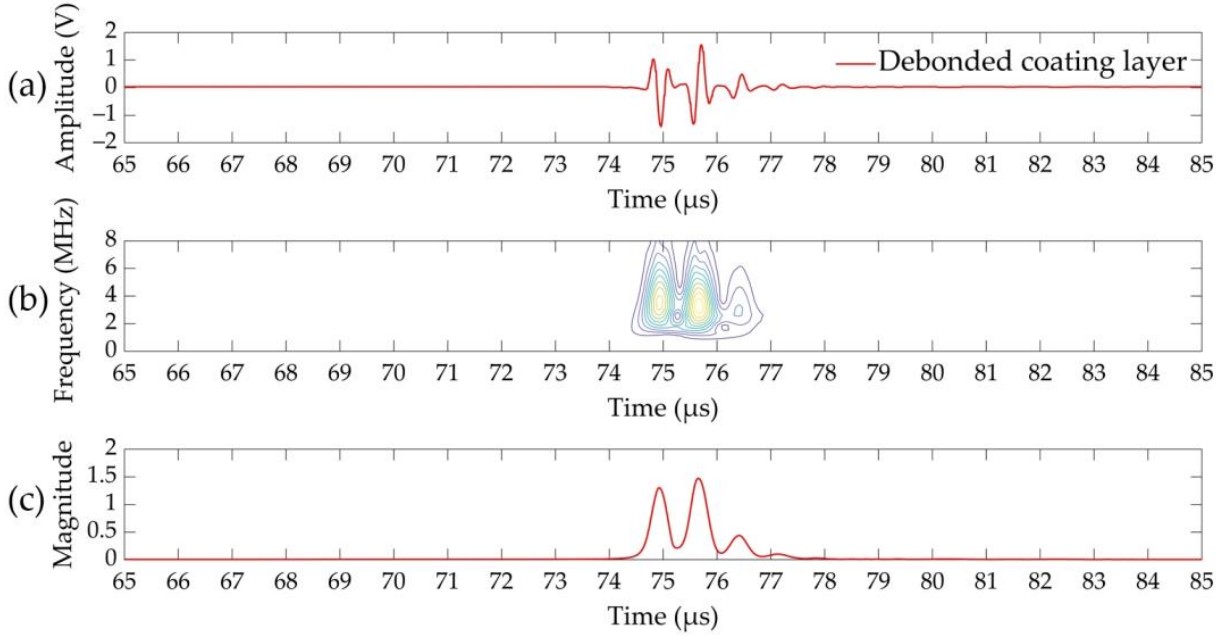

**Figure 5.** Pulse echo waveform from the debonded coating layer: (**a**) time domain waveform, (**b**) CWT of the waveform, and (**c**) the projection of the CWT magnitude to the time domain.

**Table 2.** Waveform parameters from bonded and debonded layers.

| TOF of A0 (μs) | Phase of A0 (°) | CF of A0 (MHz) | TOF of A1 (μs) | Phase of A1 (°) | CF of A1 (MHz) |
|---|---|---|---|---|---|
| **Debonded Section of Coated Concrete Specimen** | | | | | |
| 74.93 | 90 | 3.58 | 75.65 | −90 | 3.12 |
| **Bonded section of coated concrete specimen** | | | | | |
| 75.16 | 90 | 3.58 | 75.97 | 90 | 3.50 |

Another important parameter is the phase of the reflected echo, so in this research, the phases of the waveforms at the local peaks of the CWT were calculated. The advantage of the calculation of the phase in a single point is the reduction in the computational effort. The waveform can also be analyzed in the time-phase domain, but this method does not immediately highlight the features of the waveform. The Hilbert transform, together with CWT parameters (as described in Equation (7)), was used to calculate the phase of the reflected echoes. The phases of the echoes at the local peaks of CWT are shown in Table 2, and they are denoted as the "Phase of A0" for echo A0 and the "Phase of A1" for echo A1. The values of the phases at the local peak of the CWT are similar for A0 echoes, regardless of the bonding state of the coating layer. This is because the phase of the A0 echo depends only on the acoustic impedance difference between the coating layer and the water. On

the other hand, the values of the phase of the A1 echoes are different, depending on the bonding condition of the coating layer. The phase of the reflected echo from the backwall of the coating layer is reversed due to the fact that the debonded space between the water and the concrete will be occupied with either water or air. Water and air have a lower acoustic impedance than concrete and coating materials, resulting in negative reflection. HT and CWT (Equation (7)) were used to calculate the phase of reflected echoes. Table 2 displays the phase values of the echoes, showing that the phase of the A1 echo was shifted by 180 degrees for the debonded case compared to A0 based on the numerical estimates.

The local peaks of the projection of the CWT magnitude correspond to the TOF waveform. The TOF of the reflected echo was derived by projecting the CWT onto the time domain, as shown in Figure 5c. Estimated values of the TOF of the wave echoes of A0 and A1 are shown in Table 2, and they are denoted, respectively, as "TOF of A0" and "TOF of A1". Therefore, if the wave propagation speed in the coating layer is known, the thickness can be estimated. Alternatively, the TOF of reflected echoes can be calculated manually from the time domain waveform (Figure 4); however, doing so is a time-consuming process. Fast inspection is crucial when inspecting large specimens. Based on the TOFs of echoes, the layer's thickness is estimated to be 0.86 mm. Additionally, another advantage of evaluation of the TOF of the wave echoes based on the local peaks of the CWT magnitude is the capability to obtain precise results under the low signal-to-noise ratios, which is very convenient in order to detect coating layer wall flattening. As the walls of the coating layer become thinner, the echoes are superimposed in the time domain, and in those cases, the evaluation of TOF based on the CWT scalogram becomes necessary.

The bonded section was also inspected in the same way, and the corresponding waveform and its CWT are shown in Figure 6. The time-domain waveform is shown Figure 6a,b shows the time-frequency domain representation of the waveform. Similarly, for the debonded case, the instantaneous center frequencies of the reflected echoes for the bonding layer were estimated using Figure 6b, and the results are shown in Table 2. In addition, the echo phase of the local peak of the CWT magnitude of the bonded coating material was also calculated using the HT and CWT wave parameters. The calculated values are also listed in Table 2. Correspondingly, the TOF of the A0 and A1 echoes were calculated based on the local peaks of the projection of the CWT scale graph to the time domain as shown in Figure 6c, and the values can be seen in Table 2. The thickness of the coating layer was measured using the TOF, and it was found to be 0.97 mm thick.

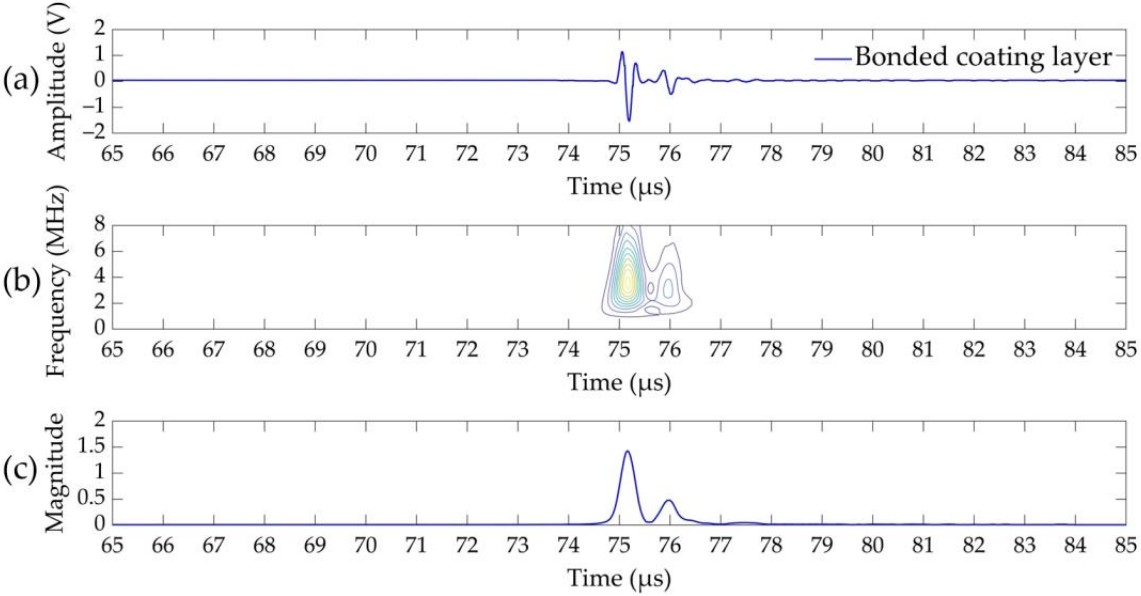

**Figure 6.** Time-domain waveform from bonded coating layer (**a**), the CWT of the time domain waveform (**b**), and the projection of the CWT magnitude to the time domain (**c**).

Additionally, the TOF of the echo from the surface A0 and that from the backwall of the coating layer A1 were slightly different. The TOF of the A0 echo is related to the distance between the ultrasonic probe and the coating layer surface, and the nonuniform thickness of the concrete affects the TOF of A0 echoes. In addition, the difference in the TOF of the A1 echo can be explained by a slightly different coating material thickness due to the application method. Additionally, the bonded coating layer has a lower CWT magnitude of A1 echo compared to the debonded layer due to the wave energy transformation from coating to the concrete. In the debonded layer, the trapped air between the coating layer and the concrete causes the reflection of nearly all ultrasonic energy, thus increasing the CWT magnitude of A1.

Based on the results of the experiments, it is very difficult to establish a certain threshold value for classifying the state of the coating layer. In this study, we introduce three more relative parameters to classify the bonding state of the coating material. The first parameter is the ratio of the local peaks of the CWT magnitude of the echoes reflected from the back wall and the surface of the coating material. The ratio of the local peaks of the CWT magnitude determines the attenuation rate of wave energy in the coating material itself. The second parameter is the phase difference between the A1 and A0 echoes, since, as described above, the phase of the echo depends on the bonding condition between the two media. Last but not least, the ratio of the instantaneous center frequencies of the A0 and A1 echoes is a unitless parameter that compares the scattering of the reflected echoes from the surface and the back wall of the coating layer. The calculated relative parameters for both bonded and detached layer are shown in Table 3 below.

**Table 3.** Waveform unitless parameters obtained from CWT and Hilbert transforms.

| Debonded Section of Coated Concrete Specimen | | |
|---|---|---|
| **CWT Ratio of A1/A0 (Unitless)** | **Phase Difference (°)** | **CF Ratio of A1/A0 (Unitless)** |
| 1.13 | 180 | 0.87 |
| **Bonded section of coated concrete specimen** | | |
| 0.27 | 0 | 0.98 |

In real applications, the local density of the coating material can vary due to the presence of inhomogeneous porosity in the coating layer, and the water quality can also change due to environmental influences. It is difficult to set threshold values for each parameter, since the values may change depending on the instrument settings, water condition, and coating layer porosity. In this study, three relative parameters of the echoes were selected, and these values were used as the input of the DNN. The details of the DNN are described in the next section below.

## 5. Neural Networks

### 5.1. Introduction to Neural Networks

The benefits of NN are its ease of design and extremely strong computational power. Furthermore, NN-based methods are less expensive than similar classification systems. NNs are predictable in the absence of knowledge of the system's principles and models. This is accomplished by training the NN using available data. NNs can learn from previous results and predict new situations. If sufficient data are available, NNs enable expert system solutions. Neural networks are made up of interconnected elements known as nodes or "neurons". To provide a dynamic data flow, neurons are layered. The first NN layer is an input layer with one neuron for each waveform input. Neurons are arranged in three or more layers, including an input layer, one or more hidden layers, and an output layer. One of the most important steps in determining useful information representing the state of the system is feature extraction from raw signals.

*5.2. Hyperparameters Optimization*

To achieve adequate performance, hyperparameters must be adjusted until optimal values are obtained [37,84,85]. A neural network's hyperparameters are a set of parameters, namely the number of neurons, the activation function, the optimizer function, the learning rate, the batch size, and the epochs. Kernel size, padding values, stride values, and the selection of the activation function can be used to tune the training process and testing process of the NN. There are several approaches and methods in the literature for tuning hyperparameters. In some cases, using a single algorithm may not produce satisfactory results. Setting hyperparameters can optimize computational resources, which is critical when building portable immersion ultrasound scanning devices with neural networks. In this study, the Bayesian optimization method was used for optimal tuning of neural network hyperparameters. The Bayesian optimization principle is to find the maximum of the function at the sampling points [38,86]. The ranges of hyperparameters and their corresponding ranges are shown in Table 4.

**Table 4.** Neural networks hyperparameter ranges.

| Hyperparameter | Range |
| --- | --- |
| Neurons | [3:100] |
| Learning rate | [0.01:1] |
| Batch size | [5:100] |
| Epochs | [20:200] |
| Number of Hidden Layers | [1:5] |
| Dropout rate | [0:0.3] |
| Optimization function | SGD, Adam, RMSprop, Adadelta, Adagrad, Adamax, Adam |
| Activation function | ReLu, Sigmoid, SoftPlus, SoftSign, Tanh, Selu, Elu, Exponential |

The batch size is the number of subsamples used to train the NN. Smaller batch sizes result in lower computational margins and faster computations, but they also increase the variance in the test accuracy. Another parameter is the optimizer selection, which influences the learning rate and neuron weights. Choosing the best optimizer can improve accuracy while decreasing NN losses. We used nine different optimization functions in the optimization procedure, as shown in Table 4, each with its own algorithm. It is also necessary to fine-tune the NN's learning rate. The lower the NN's learning rate and the more memory resources it requires, the more likely it is to achieve a low-loss function. The number of neurons in the hidden layer is the same in all hidden layers, but the number of neurons in the input layer is determined by the number of input features. The number of neurons in the output layer is related to the number of classes, and in binary classification, the number of neurons in the output layer is only one. However, the number of neurons in the hidden layer can be adjusted depending on the classification's complexity. Another NN parameter is the number of epochs or iterations in the network, which is used to determine the complexity of the classification problem. When the number of epochs is small, the NN underfits, and as the number of epochs increases, the NN overfits. Underfitting and overfitting are both undesirable phenomena in NNs; underfitting refers to a lack of learning, while overfitting refers to erroneous predictions on the test set. The dropout layer, which is responsible for deleting unnecessary neurons from the hidden layer, is another important layer. The dropout rate is typically very low, and its value defines the percentage of neurons that were dropped, because dropped neurons are no longer used empirically.

Bayesian optimization with 40 initial steps and 5random iterations yielded the best value for the hyperparameters, as shown in Table 5. The NN was built using the Bayesian optimization approach, and its schematic architecture is shown in Figure 7. The hyperparameter value was used in the NN's subsequent prediction.

**Table 5.** List of the optimal hyperparameters computed by the Bayesian approach.

| Hyperparameter | Optimal Value |
|---|---|
| Neurons in each layer | 11 |
| Learning rate | 0.08 |
| Batch size | 5 |
| Epochs | 60 |
| Number of hidden layers | 5 |
| Dropout rate | 0.01 |
| Optimization function | Adadelta |
| Activation function | ReLu |

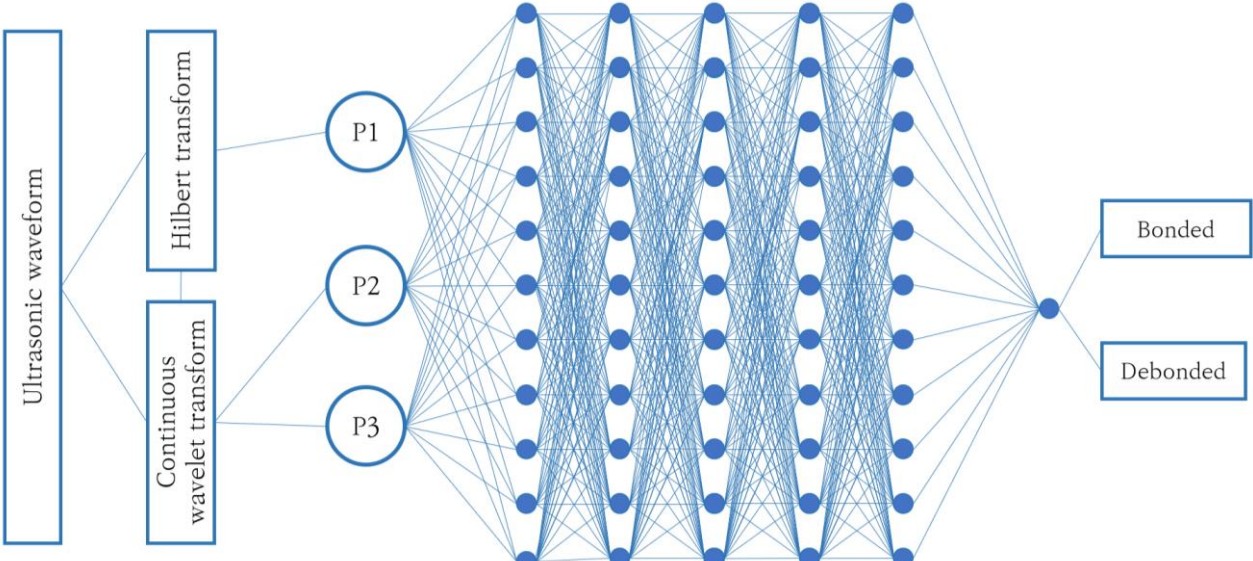

**Figure 7.** The DNN structure for the automatic classification of the bonding condition of the coating layer.

*5.3. Classification Results*

Based on the optimum results of hyperparameters, deep neural networks (DNNs) were designed, and the schematic representation of DNN is shown in Figure 7 below. We have implemented the proposed DNN model by using the Python language and the Spyder platform. DNN refers to the presence of several hidden layers between the input and output layers of the NN, and in our case, the optimized model consists of five hidden layers, with 11 neurons in each layer. Initially, the CWT was applied to the waveform and relative parameters such as the instantaneous center frequency ratio and the CWT magnitude ratio were calculated. The phase difference between echoes was calculated by applying the HT to the waveform and using measured instantaneous center frequency and the TOF by CWT. The computed relative parameters were used to train DNN. As shown in Figure 7, the DNN structure is schematically represented by the input relative parameters P1, P2, and P3, which, respectively, correspond to phase difference, instantaneous center frequency ratio, and CWT magnitude ratio.

In this research, the number of specimens was fabricated using the method described in Section 3.1, and ultrasonic measurements were obtained. The dimensions of the training, validation, and test set are shown in Table 6. Additionally, classification results were plotted as a confusion matrix, as shown in Figure 8.

**Table 6.** The dimension of the training, validation, and test set of DNN.

| Training Set | Validation Set | Test Set | Total |
|:---:|:---:|:---:|:---:|
| 638 | 160 | 794 | 1592 |

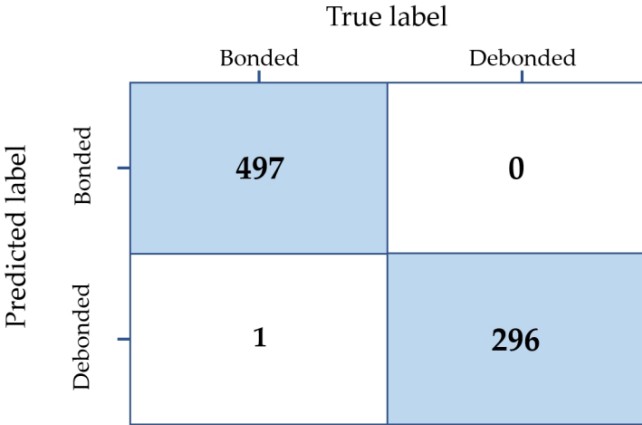

**Figure 8.** Confusion matrix of the classification results.

As shown in the confusion matrix of the DNN classification, the accuracy of the classification for the debonded layer was 100%. This proves that the selected three relative parameters of the waveform can predict the bonding condition of the coating layer. Additionally, the DNN with the optimized parameters showed high accuracy in the classification of the bonding state of the coating layer.

## 6. Discussion

In this research, we used waveform parameters to the classify bonding state of the coating material. In particular, the ratio of local peaks on the CWT magnitude, which is proportional to the echo amplitude, is a relative parameter to compare the attenuation rates in different coating layers. As shown in Figures 5 and 6, the CWT magnitude of the A1 echo for bonded coating layers was lower compared to the debonded coating layers. Due to the wave energy transmission between the coating layer and concrete, bonded layers attenuate at a faster rate than debonded layers [87,88].

The phase of the echoes computed by CWT and HT of the waveform confirms the theoretically derived derivation described in Section 2. The phase of the echo from the surface and the back wall concrete was equal when the coating layer was bonded to the concrete substrate. As a matter of fact, both acoustic energy reflection coefficients in the bonded case were positive [89]. In the case of delamination, the space between the coating and the concrete is occupied with water or air, leading to a negative reflection coefficient (as seen in Table 1), and the phase of the second echo is reversed by 180° [90]. The method would be accurate even if seawater instead of tap water were used during the experiments, since the acoustic impedance difference between tap water and seawater is not significant. These values are $1.4 \times 10^6$ kg/(m²s) and $1.53 \times 10^6$ kg/(m²s), respectively.

The waveform parameters calculated in Table 2 depend on instrument setting, water condition, and coating layer thickness [91]. For instance, variations in water salinity affect the amplitude of reflected echoes because of acoustic impedance effects. A probe's frequency and bandwidth parameters can also affect the received waveform. As frequency increases, the attenuation rates increase, and the narrow band parameters can result in echoes overlapping. In this study, three relative parameters of the wave echoes were used to classify the bonding condition of the coating and to eliminate the effect of variable parameters of the experiment. As can be seen from Table 3, the CWT ratio was greater in the bonded layer, which indicates a higher attenuation rate. Another relative parameter is the phase difference between echoes, which is zero when the coating layer is bonded.

A delaminated layer has a phase difference of 180° [92–94]. Additionally, the CF ratio of reflected echoes was used as a parameter to classify the bonding state. A higher CF ratio between A1 and A0 indicates more scattering of wave energy between the coating layer and concrete substrate [95,96].

Both the accuracy and the precision of the prediction are hyperparameters of the DNN effects. Using DNN with optimally tuned hyperparameters, we were able to automatically classify the bonding state of coating layers. Three relative parameters of the reflected echo were sensitive to delamination, allowing 100% accuracy in detecting debonding.

## 7. Conclusions

In this study, a nondestructive immersion ultrasound method was used to detect delamination of the coating layer in concrete structures. The possibility of detecting debonding layers was confirmed through continuous wavelet transforms using complex Morlet wavelets together with Hilbert transform. The results of this study are summarized as follows.

1.  The local peaks of the CWT magnitude were used to estimate the TOF of the reflected echoes. Despite the fact that the TOF of a reflected echo can also be determined manually in a time-domain waveform, the CWT-based method has the advantage of estimating the TOF of very thin coating layers regardless of water quality, even if the signal-to-noise ratios are low.
2.  The CWT magnitude ratio of the two consecutive echoes has been used as a relative parameter to compare attenuation rates in the coating layer. The attenuation rate of the ultrasonic energy was faster in the bonded compared with the delaminated coating layer due to the ultrasonic energy transfer to the concrete. Consequently, the CWT magnitude ratio was higher in the debonded layer.
3.  The phase at local peaks of CWT magnitude showed a clear difference between bonded and delaminated coating layers. The phase of the reflected echo was shifted by 180 degrees when the coating was delaminated from the concrete.
4.  Automatic classification of the bonding condition was achieved by DNN through the selection of optimal hyperparameters using the Bayesian method. A total of 1592 data sets were used to train, test, and validate the DNN's performance. The design model predicts the delamination pattern without error for all validation specimens. DNN made an automatic decision to inspect the bonding state without setting any threshold values for detection.
5.  The developed method is robust and allows for simultaneous inspection of coating layer thickness and adhesion. By incorporating computed relative parameters and DNN, it is possible to examine the bonding state of the coating material even if the distance between the probe and the specimen is changed.

**Author Contributions:** Conceptualization A.K.u.M. and Y.H.K.; methodology, A.K.u.M. and Y.H.K.; software, A.K.u.M. and Y.H.K.; validation, A.K.u.M. and Y.H.K.; formal analysis, A.K.u.M. and J.K.; investigation, Y.H.K. and J.-H.Y.; resources, J.-H.Y.; data curation, A.K.u.M.; writing—original draft preparation, A.K.u.M.; writing—review and editing, Y.H.K. and Y.C.; visualization, A.K.u.M.; supervision, Y.C.; project administration, J.-H.Y. and J.Z.; funding acquisition, J.-H.Y. and J.Z. All authors have read and agreed to the published version of the manuscript.

**Funding:** The authors appreciate being supported by the National Research Foundation of Korea (NRF) grant funded by the Korean government (MSIP) (No. 2020M2D2A1A02069933).

**Institutional Review Board Statement:** Not applicable.

**Informed Consent Statement:** Not applicable.

**Data Availability Statement:** Not applicable.

**Acknowledgments:** We are highly grateful to Sung-Un Lee for helping with the English grammar correction.

**Conflicts of Interest:** The authors declare no conflict of interest.

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
