# Peer review of "Neural-Network-Based Ultrasonic Inspection of Offshore Coated Concrete Specimens"

_coatings, doi:10.3390/coatings12060773_

Round 1

Reviewer 1 Report

The research is interesting. There are some suggestions to further improve the paper.

  1. Introduction; few statements about concrete in the first paragraph are not very accurate. Yes, concrete has porosity but not to say it has “porous structure”. High quality concrete has very low porosity. If porosities are not interconnected then there will not be major issues in an aggressive environment and under water. However, microstructure of concrete surface may differ from inside the concrete. Please rewrite this part and use reference to support the important statements.
  2. Introduction; second paragraph; elaborate more “bonding between the concrete surface and coating materials”. Please use references to support the statements.
  3. Line 88; give reference numbers after the statement.
  4. Line 193; basic properties of concrete substrate should be presented here as it is important in analysis. Please do not force the readers to study another paper just to get basic and important data related to this study.
  5. The research methodology clearly mentions what were the variables in this research.
  6. Line 204; explain more about the polyethylene layer. Is it common to be used also in practice? How much was the thickness of the coating of material? How much is the common thickness of these two layers in practice?  
  7. Section 3.2; the water used was normal tap water? Please specify.  
  8. Results are well presented but there is no reference to support the justifications, making discussion to be poor.  

Author Response

Response to Reviewer 1 Comments

We appreciate the time and effort that you and the reviewers dedicated to providing feedback on our manuscript and are grateful for the insightful comments on and valuable improvements to our paper. We have incorporated most of the suggestions made by the reviewers. Those changes are highlighted within the manuscript. Please see below, in blue, for a point-by-point response to the reviewers’ comments and concerns. All page numbers refer to the revised manuscript file with tracked changes.

  1. Introduction; few statements about concrete in the first paragraph are not very accurate. Yes, concrete has porosity but not to say it has “porous structure”. High quality concrete has very low porosity. If porosities are not interconnected then there will not be major issues in an aggressive environment and under water. However, microstructure of the concrete surface may differ from inside the concrete. Please rewrite this part and use reference to support the important statements.

Answer: Thank you so much for your suggestion. The authors completely agree with the comment above. The degradation is time-dependent and cause by chemical and physical phenomena. Corrosion initiation depends on the marine environment.  The first part was rewritten and references were included. Hopefully, you will find it logical.

  1. Introduction; second paragraph; elaborate more “bonding between the concrete surface and coating materials”. Please use references to support the statements.

Answer: Thank you so much for your suggestion.  Detailed information regarding the causes of the defect was added to the second paragraph, as well as references relating to the bonding of the coating to the concrete. We have mentioned it in the revised manuscript.

  1. Line 88; give reference numbers after the statement.

Answer: Thank you so much for your comment. A literature review was added regarding “detection of the coating layer applied to the concrete structure in the water” in the revised manuscript. Hopefully, you will find justified.

  1. Line 193; basic properties of the concrete substrate should be presented here as it is important in analysis. Please do not force the readers to study another paper just to get basic and important data related to this study.

Answer: Thank You for the correction. The main properties of the concrete substrate regarding ultrasonic testing were added.  Hopefully, you will find it justified.

  1. The research methodology mentions what were the variables in this research.

Answer: Thank you so much for your comments.  In Subsection 2.2 the range of variables was included and the index was clarified. Hopefully, you will find it justified.

  1. Line 204; explain more about the polyethylene layer. Is it common to be used also in practice? How much was the thickness of the coating of the material? How much is the common thickness of these two layers in practice?  

Answer: Thank you so much for your useful comment, answering them we improved the quality of the manuscript.

  1. Polyethylene layers are commonly used to simulate a debonded layer. To create the debonded section of the layer, non-adhesive material must be used. Here is the reference [1]  [2]  
  2. The general thickness of the the coating layer might be thin and rich up to 3 mm.
  3. In the experiments the main goal was not to design the optimal thickness of the coating layer, therefore we showed two cases with different thicknesses. The thickness of the debonded layer was 0.85 mm whereas the bonded is 1.7 mm    

Hopefully you will find it logical.

  1. Section 3.2; Is the water used as normal tap water? Please specify. 

Answer:   Thank you so much for your comments.  In the experimental studies, the tapered water was used in the immersion studies, whereas in a real application the offshore structure is located in seawater. The seawater contains possesses a higher concentration of the salinity and sediment content compared with tap water. As result, the seawater processes higher density, which leads to lower wave propagation speed and faster attenuation rate of the ultrasonic energy [3] [4]. With the attenuation of ultrasonic energy in the coating material, the difference TOF echoes from the surface and back wall of the coating layer do not affect by water salinity and sediment content. Also, the phase change reflected is independently affected by the salinity of the water. Hopefully you will find it justified.

  1. Results are well presented but there is no reference to support the justifications, making the discussion to be poor.  

Answer: Thank you so much for your comments.  The Discussion part was added in the manuscript, where all results were referred to with references (Please refer to Section 6 ). We have added it to the revised manuscript.

REFERENCES:

  1. Wang, B.; Shi, W.; Zhao, B.; Tan, J. A modal decomposition imaging algorithm for ultrasonic detection of delamination defects in carbon fiber composite plates using air-coupled Lamb waves. Measurement 2022, 195, 111165, doi:10.1016/j.measurement.2022.111165.
  2. Smagulova, D.; Jasiuniene, E. Non - destructive evaluation of dissimilar material joints. 1–4.
  3. Fortineau, J.P.; Vander Meulen, F.; Fortineau, J.; Feuillard, G. Efficient algorithm for discrimination of overlapping ultrasonic echoes. Ultrasonics 2017, 73, 253–261, doi:10.1016/j.ultras.2016.09.010.
  4. Wang, J.; Liu, B.; Kan, G.; Li, G.; Zheng, J.; Meng, X. Frequency dependence of sound speed and attenuation in fine-grained sediments from 25 to 250 kHz based on a probe method. Ocean Engineering 2018, 160, 45–53, doi:10.1016/j.oceaneng.2018.04.078.

Reviewer 2 Report

This paper introduces the application of a type of neural networks based on ultrasonic approach, which aims to evaluate the offshore coated concrete. As a kind of NDT method, acoustic approach is a relatively fully non-destructed way which is novel enough. The manuscript is well constructed and edited, which could be considered to be published after major revision. And also, I think the English editing is high qualified to be published. I just have several recommendations as follows:

  1. The authors applied reflection/echo type of acoustic method, how about the penetration type of acoustic method to detect the fracture, which means two probes were used……
  2. 4: It seems like the received waveform of bonded and de-bonded coating layer does not differ a lot, please explain in details how to distinguish them two?
  3. The authors should carry out some microscopic tests to prove that the existence of fractures/peeling indeed causes the attenuation of reflected waves.

Overall, the authors are highly encouraged to resubmit the revised manuscript to us again.

Author Response

Response to Reviewer 2 Comments

Coauthors and I greatly appreciate the reviewer's encouraging, critical and constructive comments on this manuscript. The comments have been very thorough and helpful. We strongly believe that the comments and suggestions have increased the scientific value of the revised manuscript many folds. These comments and suggestions have been fully incorporated into the revised manuscript. The corrected manuscript is being submitted with the suggestions incorporated. The manuscript has been revised as per the comments given by the reviewer, and our responses to all the comments are as below.

  1. The author's applied the reflection/echo type of acoustic method, how about the penetration type of acoustic method to detect the fracture, which means two probes were used……

Answer:  Thank you so much for your comment. As it was mentioned correctly, in the current research ultrasonic pulse-echo was implemented. The ultrasonic NDT method can be done also by penetration method, however, it causes the following inspection difficulties:

  1. In a real application, the concrete wall thickness might reach up to several meters, which makes it almost impossible to transmit the ultrasonic energy through the structure and receive be a receiver. Based on experience, the concrete was inspected mainly at low frequencies (below 100 kHz), and these low-frequency waves were simply not sensitive to the coating layer's presence.
  2. In the immersion ultrasonic inspection, it is required to have a compact (portable) ultrasonic scanning instrument. Adding the second probe (ultrasonic probe increases size).
  3. Last but not least, implementation of the acoustic penetration method might require two scanning operators. Whereas in the case of the pulse-echo method testing can be performed by a single operator.

Hopefully, you will find it logical.  

  1. 4: It seems like the received waveform of bonded and de-bonded coating layer does not differ a lot, please explain in detail how to distinguish the two?

Answer: Thank you so much for your comments. Yes, we completely agree that it is very difficult to differentiate between the bonded state condition from the time domain waveform. In contrast, CWT can be used to compute the attenuation rate and the phase of reflected echoes, whereas HT can be used to determine the phase of reflected waves. In the current research, applied CWT and HT the allowed to compute relative parameters of the waveform which is very good in classification coating layer bonding state by DNN.

 According to theory, the debonded case exhibits a lower attenuation rate than the bonded case. Because in the bonded layer wave energy is transmitted to the concrete.

Additionally, the phase of the echo will be reversed when the coating layer is debonded, due to the acoustic impedance difference between the coating layer and concrete.

Corresponding notes is added to the revised manuscript. 

  1. The authors should carry out some microscopic tests to prove that the existence of fractures/peeling indeed causes the attenuation of reflected waves.

Answer: Thank you so much for your precious suggestion. In the Ultrasonic nondestructive testing, it is very common to use polyethylene layer coating material and concrete because the polyethylene layer is nonadhesive, and very thin (thickness of the layer is less than the wavelength) and acoustic impedance is similar to the coating layer. Corresponding texts were added to the revised manuscript and the following reference were added [1]  [2] in the revised manuscript.

References:

  1. Wang, B.; Shi, W.; Zhao, B.; Tan, J. A modal decomposition imaging algorithm for ultrasonic detection of delamination defects in carbon fiber composite plates using air-coupled Lamb waves. Measurement 2022, 195, 111165, doi:10.1016/j.measurement.2022.111165.
  2. Smagulova, D.; Jasiuniene, E. Non - destructive evaluation of dissimilar material joints. 1–4.

Reviewer 3 Report

General Comment

The manuscript presents an experimental and numerical study which aims to propose a methodology, based on neural networks, to detect debonding of the coating layer of offshore concrete elements through immersion ultrasound. For this, experimental concrete samples were produced in the laboratory, with bonded and unbonded coating, and then immersed in water. Ultrasonic immersion tests were performed and the results are presented and discussed. Measured wave parameters were used to train a deep neural networks, which was then used to predict the coating layer’s bonding state. It was found that the proposed methodology is able to predict with accuracy both the coating layer’s bonding state and thickness.

The topic of the manuscript is very interesting since it deals with immersed offshore concrete elements for which the life time strongly depends on the durability of concrete. The evaluation of the coating layer state of such elements, under immersion condition, can be very hard. The proposed methodology in the manuscript constitutes a good contribution to help solving this drawback and could be useful for future researches and also for practice.

I made some comments in order to improve the manuscript. The authors should take the comments into account and revise their manuscript.

Specific Comment 1

Please revise the entire manuscript to improve the reading and correct typos. Several sentences must be rewritten for better understanding. The manuscript should be revised by a native English speaker. I don’t present examples because they are too many!

Specific Comment 2

Introduction

In the literature review, the authors should refer previous studies related with the specific topic of the manuscript, i.e., “researches on the detection of the debonded section of the coating layer applied to the concrete structure in water immersion” (as referred by the authors at lines 88-89) by using DNN. The main results of such studies should be summarized and discussed. Also, the authors should clarify what is the novelty of their study when compared with the previous ones.

Specific Comment 3

Section 3.2

The experimental samples were submersed in simple water. How the obtained results can be extrapolated to real offshore concrete elements submersed in sea water? Please refer and discuss this in the manuscript.

Specific Comment 4

The title of Section 5 and Section 5.1 should not be the same

Specific Comment 5

Section 5

How the DNN model was implemented in the computer? Did the authors use a commercial software or did they implement a code in a programing language? Please give more information on this in the manuscript.

Specific Comment 6

Section 6

The conclusions section is poor and should be improved.

Author Response

       Response to Reviewer 3 Comments

The authors would like to thank the reviewers for their precious time and useful comments. We have carefully addressed all the comments. The corresponding changes and refinements made in the revised paper are summarized in our response below.

 Specific Comment 1

Please revise the entire manuscript to improve the reading and correct typos. Several sentences must be rewritten for better understanding. The manuscript should be revised by a native English speaker. I don’t present examples because they are too many! 

Answer: Thank you so much for your comment Yes, the notes and comments of the reviewer are correct and we respect them. As a matter of fact, the authors are not native speakers, it was a bit time-consuming to correct all the manuscripts. The sentences were corrected by doing all possible efforts. In case of necessity, we will apply for a grammar correction service. Hopefully, you will find justified.

Specific Comment 2 (Introduction)

In the literature review, the authors should refer to previous studies related to the specific topic of the manuscript, i.e., “researches on the detection of the debonded section of the coating layer applied to the concrete structure in water immersion” (as referred by the authors at lines 88-89) by using DNN. The main results of such studies should be summarized and discussed. Also, the authors should clarify what is the novelty of their study when compared with the previous ones.

Answer: Thank you so much for your comment. A literature review was added regarding “detection of the coating layer applied to the concrete structure in the water” in the revised manuscript. Also, sentences was added in the Introduction by clarifying the novelty of the current research. Hopefully, you will find justified. 

Specific Comment 3 (Section 3.2)

The experimental samples were submersed in simple water. How the obtained results can be extrapolated to real offshore concrete elements submerged in seawater? Please refer to and discuss this in the manuscript

Answer:  Thank you so much for your comment. In the experimental studies, the tapered water was used in the immersion studies, whereas in a real application the offshore structure is located in seawater. The seawater contains possesses a higher concentration of the salinity and sediment content compared with tap water. As result, the seawater processes higher density, which leads to lower wave propagation speed and faster attenuation rate of the ultrasonic energy [1] [2]. As a matter of fact, the attenuation of ultrasonic energy in the coating material, the difference in TOF echoes from the surface and back wall of the coating layer does not affect by water salinity and sediment content. Also, the phase change reflected is independently affected by the salinity of the water. Corresponding notes was added in the manuscript.  Hopefully, now you will find it rational.

Specific Comment 4

The title of Section 5 and Section 5.1 should not be the same 

Answer:  Yes, Thank You! The comment was corrected, in the revised manuscript.

Specific Comment 5 (Section 5)

How the DNN model was implemented in the computer? Did the authors use commercial software or did they implement a code in a programing language? Please give more information on this in the manuscript.

Answer: Thank You for your comment, it was really useful. During designing the DNN Python commercial software selection was used. However, the DNN can be designed in much commercial software, such as C+ Java, Matlab, and Python. Details of the Software and corresponding libraries were included in the manuscript. We have addressed it during the revision of the manuscript.

Specific Comment 6 (Section 6)                                                            

The conclusions section is poor and should be improved.

Answer: Thank you so much for your suggestion. The conclusion section is also revised for better interpretation.

Reviewer 4 Report

SUMMARY

An article is presented for review, the topic of which is relevant today. The study involves the application of a protective coating to the surface of offshore concrete structures and its control using ultrasonic methods and using neural networks. This topic of research is very relevant in modern science and practice. Artificial intelligence methods, as well as non-destructive control methods, are one of the key areas of assessment and quality control of building structures, including hydraulic engineering.

The authors used an interesting methodological apparatus, which is based on the most modern research methods. In addition, they obtained a number of very important practical results. The analytical component of the article is also quite strong. In this regard, we can note the high originality of the study, a certain scientific novelty and practical significance. The results obtained by the authors can be applied in practice, and the theoretical ideas obtained by them for scientific purposes can also be developed.

However, the article also has some shortcomings. They are listed below by the reviewer.

COMMENTS

  1. The Abstract contains rather vague achievements that should be specified. The authors operate with forms of phrases like "with high accuracy", as well as some other qualitative characteristics of any processes and objects, but do not give a quantitative description of these processes and objects. The Abstract should be supplemented to better reflect the content of the article.
  2. The "Introduction" section contains a literature review on the research topic, however, a rather small number of sources analyzed by the authors attracts attention. Such actual topics as artificial intelligence, ultrasonic control methods and hydraulic structures, of course, have great elaboration and the authors should analyze another 10-15 sources in key areas of research.
  3. It is necessary to add specifically formulated goals and objectives of the study to the "Introduction" in order to clearly express the scientific novelty and practical significance.
  4. In section 2, it is necessary to add a program of experimental and theoretical studies to understand the structure of the work, since this section begins immediately with a description of materials and methods.
  5. The sentences on lines 102-106 require additional justification. In particular, the authors state that usually non-destructive testing methods are based on the estimation of the variation in the parameters of the ultrasonic wave of reflected echo signals, and describe this process. A reference to the original source should be added here, since this issue is rather not scientific, but practical, and it is largely regulated by norms.
  6. It should be noted that subsection 2.1 ends with Figure 1. This probably reduces the degree of perception of the article by the reader. In the opinion of the reviewer, a textual interpretation of the figure should be added after the figure itself and a logical transition should be made from subsection 2.1 to subsection 2.2.
  7. Subsection 2.2 contains a large number of formulas that need to be deciphered for the values ​​used. That interpretation, which is given in its current form, is insufficient due to the difficulty of perceiving the designations of the quantities used in the text, and also, perhaps, is inconvenient for the reader to understand.
  8. Subsection 2.3 also ends with a table, which should be interpreted in the text after it. This will move you from section 2 to section 3.
  9. In section 3 "Experiments", the sentences on lines 188-193 probably require additional explanations and arguments, despite the reference given.
  10. The remark at the end of the subsection with a figure also applies to subsections 3.1 and 3.2.
  11. Subsection 3.3 ends with a table that should be interpreted in the text.
  12. Table 2 is not very informative due to the small amount of data presented in this table. Probably, such a representation would be more convenient in the form of text, but this is at the discretion of the authors.
  13. The first mention in the text of Figure 4 must be placed before the figure.
  14. Figure 4 needs some additional interpretation, if only for 1 or 2 sentences. The same applies to Figures 5 and 6.
  15. Unfortunately, the article does not have a "Discussion" section. This is unacceptable and it is necessary to add a “Discussion” section after section 5, in which a detailed comparison of the results obtained by the authors of the article with the results obtained earlier by other authors should be added.
  16. The Conclusions are succinct and perhaps they should have been given in a slightly more detailed form, substantiating the results obtained in more detail.
  17. The list of references should be analyzed in terms of scientific novelty, since there are quite a lot of sources older than 5 years. In order to talk about the scientific novelty and prospects of the study, it would be necessary to cite a larger number of fresh sources over the past 5 years.

Author Response

Response to Reviewer 4 Comments

We thank the reviewer for their valuable comments. These comments are very constructive and will help us to improve our manuscript, particularly in terms of clarifying our methodology and our goal. We have responded to the reviewer's concerns in this letter, and the manuscript will be improved as a result.

  1. The Abstract contains rather vague achievements that should be specified. The authors operate with forms of phrases like "with high accuracy", as well as some other qualitative characteristics of any processes and objects, but do not give a quantitative description of these processes and objects. The Abstract should be supplemented to better reflect the content of the article.

Thank you so much for your suggestion. The detail regarding the accuracy was included in the revised manuscript. Hopefully, you will find justified Hopefully, you will find justified Hopefully, you will find justified

  1. The "Introduction" section contains a literature review on the research topic, however, a rather small number of sources analyzed by the authors attracts attention. Such actual topics as artificial intelligence, ultrasonic control methods, and hydraulic structures, of course, have great elaboration and the authors should analyze another 10-15 sources in key areas of research.

Answer: Thank you so much for your comment. In the introduction part of the manuscript corresponding reference literature was included.

  1. It is necessary to add specifically formulated goals and objectives of the study to the "Introduction" in order to clearly express the scientific novelty and practical significance.

Answer: Thank you so much, the comment is very useful.  In the “Introduction” in the last section, the statement about novelty was clarified. Please refer to the revised manuscript.

  1. In section 2, it is necessary to add a program of experimental and theoretical studies to understand the structure of the work, since this section begins immediately with a description of materials and methods.

Thank You for your comments, we agree with your comment. The brief outlines of theoretical and experimental works added in Section 2

  1. The sentences on lines 102-106 require additional justification. In particular, the authors state that usually non-destructive testing methods are based on the estimation of the variation in the parameters of the ultrasonic wave of reflected echo signals, and describe this process. A reference to the source should be added here, since this issue is rather not scientific, but practical, and it is largely regulated by norms.

Answer: Thank you so much for your suggestion. Corresponding references were added. We have addressed it during the revision of the manuscript.

  1. It should be noted that subsection 2.1 ends with Figure 1. This probably reduces the degree of perception of the article by the reader. In the opinion of the reviewer, a textual interpretation of the figure should be added after the figure itself and a logical transition should be made from subsection 2.1 to subsection 2.2.

Answer: Thank you so much for your comment. Corresponding interpolation text was added in Subsection 2.1.

  1. Subsection 2.2 contains a large number of formulas that need to be deciphered for the values ​​used. That interpretation, which is given in its current form, is insufficient due to the difficulty of perceiving the designations of the quantities used in the text, and also, perhaps, is inconvenient for the reader to understand.

Answer: Thank you so much for your useful comments.  Values the variables were clarified in the expressions, as can be seen in Section 2.2 . We have addressed it during the revision of the manuscript.

  1. Subsection 2.3 also ends with a table, which should be interpreted in the text after it. This will move you from section 2 to section 3.

Answer: Thank you so much for your precious suggestion. Corresponding interpolation text  of the Table was added in Subsection 2.3

  1. In section 3 "Experiments", the sentences on lines 188-193 probably require additional explanations and arguments, despite the reference given.

Answer: Thank you so much for your important comment. The main properties of the concrete substrate regarding ultrasonic testing were added.

  1. The remark at the end of the subsection with a figure also applies to subsections 3.1 and 3.2.

 Answer: Thank you so much for your comments. Additional information was added in the revised manuscript.  

  1. Subsection 3.3 ends with a table that should be interpreted in the text.

Answer: Thank you so much for your comments. Details were added together with corresponding references.

  1. Table 2 is not very informative due to the small amount of data presented in this table. Probably, such a representation would be more convenient in the form of text, but this is at the discretion of the authors.

Answer:  We followed Your suggestion. Table 2 was replaced with a textural explanation of the equation. Also, the geometry-related parameters of the ultrasonic probe were added. Hopefully you will find it logical

  1. The first mention in the text of Figure 4 must be placed before the figure.

Answer: Thank you so much for your comments. The position of Figure 4 in the text was shifted to the proper location

  1. Figure 4 needs some additional interpretation, if only for 1 or 2 sentences. The same applies to Figures 5 and 6.

Answer: The details of the results were added, and a corresponding description and comparison was added.

  1. Unfortunately, the article does not have a "Discussion" section. This is unacceptable and it is necessary to add a “Discussion” section after section 5, in which a detailed comparison of the results obtained by the authors of the article with the results obtained earlier by other authors should be added.

Answer: Additional section for Discussion was added, as can be seen in Section 6. Hopefully it has addressed your concern.

  1. The Conclusions are succinct and perhaps they should have been given in a slightly more detailed form, substantiating the results obtained in more detail.

Answer: Thank You very much, we agree with comments. The conclusion was rewritten, by underlining the main results. The whole section is significantly revised for better interpretation and insight. Hopefully, you will find it logical.  

  1. The list of references should be analyzed in terms of scientific novelty since there are quite a lot of sources older than 5 years. In order to talk about the scientific novelty and prospects of the study, it would be necessary to cite a larger number of fresh sources over the past 5 years.

Answer: Thank you so much for your comment. Some of the irrelevant old manuscripts were replaced with recent research works.

Round 2

Reviewer 1 Report

The authors carefully addressed  my comments. Paper is good to be published in the journal. 

Reviewer 2 Report

The paper has been revised according to the reviewer's comments. So, it can be publication by the journal.

Reviewer 3 Report

I received and read the revised version of the article “Neural networks based ultrasonic inspection of the offshore coated concrete specimens”. I´m very satisfied with the author’s replies to my comments and I also consider that all my suggestions have been properly considered by the authors to improve the article. I consider that the revised article submitted by the authors can be accepted to be published.